# Evaluating the effectiveness of applying aroma seals to masks in reducing stress caused by wearing masks: A randomized controlled trial

**Nobuyuki Wakui** [ID]*[◎], **Kotoha Ichikawa**[◎], **Aika Okami, Hinako Kagi, Shoko Kawakubo, Chikako Togawa, Raini Matsuoka, Mai Watanabe, Miho Yamamura, Shunsuke Shirozu, Yuika Tsubota, Yukiko Yoshizawa, Yoshiaki Machida**

Division of Applied Pharmaceutical Education and Research, Faculty of Pharmaceutical Sciences, Hoshi University, Shinagawa-ku, Tokyo, Japan

◎ These authors contributed equally to this work.
* n-wakui@hoshi.ac.jp

**Data Availability Statement:** All relevant data are within the paper and its Supporting Information files.

## Abstract

During the COVID-19 pandemic, face masks on patients and healthy people have been recommended to prevent airborne transmission of the virus. This increased the number of people who felt stressed while wearing masks. In this study, we investigated the stress-relieving effects of attaching aroma seals to masks. A double-blind, randomized controlled trial was conducted involving 62 university students. The participants were randomly assigned to two groups and instructed to apply a seal to their masks once a day throughout the study period. The primary measure used was the Depression, Anxiety, and Stress Scale-21 (DASS-21), while the secondary measures included the assessment of breathlessness associated with mask-wearing and the World Health Organization Five Well-Being Index (WHO-5). The intervention group, referred to as the aroma-seal use group, utilized aroma seals infused with orange-lime essential oil with the expectation of experiencing the healing effects of citrus. On the other hand, the non-intervention group, known as the placebo-seal use group, utilized identical seals without any aroma. Results indicated that the aroma-seal use group exhibited significant improvements in both the total DASS-21 scores and depression scores compared to their baseline values by the second week of the intervention. Furthermore, the aroma-seal use group demonstrated a reduced occurrence of breathlessness while wearing masks compared to the placebo-seal group. Additionally, when assessing the item "I have felt calm and relaxed" from the WHO-5 questionnaire, the aroma-seal use group displayed significantly higher scores than the placebo group. Therefore, using aroma seals containing orange–lime essential oil could be beneficial in relieving mental stress and reducing breathlessness while wearing a mask, thus improving mental health.

**Funding:** Unfunded studies Enter: The author(s) received no specific funding for this work.

**Competing interests:** NO authors have competing interests Enter: The authors have declared that no competing interests exist.

## Introduction

With the spread of the coronavirus disease 2019 (COVID-19) pandemic, people have been recommended to wear masks in various settings around the world [1]. Prior to COVID-19, most people generally used masks to prevent influenza-associated infections and pollinosis [2, 3]. Due to COVID-19, governments in many countries made indoor masking mandatory even after COVID-19 vaccines were rolled out worldwide [4–6]. As of March 13, 2023, in Japan, it has been deemed necessary to respect the individual's independent choice to wear masks indoors and outdoors. Wearing a mask is only recommended when it is effective as a countermeasure to prevent the spread of infection [7].

A recent study revealed notable disparities between Poland and China in mask-wearing and associated health outcomes. Specifically, far fewer Poles (35.0%) wore masks compared to the Chinese (96.8%), and they reported higher levels of anxiety, depression, and physical symptoms by COVID-19 [8]. These results underline cultural and regional difference in the health and stress effects of mask use.

Although masks have been shown to be effective in reducing the spread of COVID-19 [9–11], the number of people who felt stressed while wearing them also increased. In fact, more than 80% of people have experienced some type of stress, such as feeling breathlessness or feeling hot while wearing a mask [12]. Accordingly, a countermeasure to the stress caused by wearing masks is necessary.

The use of the aroma essential oils has potential to alleviate the stress caused by wearing masks. Aroma essential oils, a well-known non-invasive, natural compound [13], are considered to be effective in reducing anxiety [14], calming the mind [15, 16], and relieving stress in daily life [16–23]. Therefore, spraying an aroma essential oil on the mask or attaching an aroma seal can help relieve stress caused by wearing masks. So far, one study has reported that spraying aroma essential oils on masks reduced drowsiness and increased nurses' attention during night shifts [24]. However, spraying causes the aroma to quickly evaporate, making it difficult to achieve a long-lasting effect. Meanwhile, aroma seals are expected to be as effective at relieving stress as the aroma spray. In particular, unlike spraying, the effect of aroma seal is expected to last longer because aroma essential oil adheres to the seal. However, no scientific validation of this effect has been conducted so far, and the extent to which aroma seals alleviate stress caused by wearing masks has not been determined.

Therefore, in this study, a double-blind, randomized controlled trial was conducted to determine the stress-alleviating effects and safety of using aroma seals while wearing masks. Citrus is the one of the most popular scents in the world [25]; therefore, orange–lime essential oil, which is thought to have a refreshing effect in general, was adopted for the aroma used in the seal. This study aimed to demonstrate the usefulness of aroma seals in various situations where wearing a mask is required.

## Methods

### Study design

In this study, a double-blind, randomized controlled trial was conducted to assess the effects of aroma seals on adult participants. The study was conducted between November 11, 2021, and December 20, 2021. The participants were informed in advance about the study's objectives and methods, and written informed consent was obtained. This study was approved by the Research Ethics Committee of Hoshi University (approval no. 2021–15). The study protocol was pre-registered at the University Hospital Medical Information Network Center prior to the initiation of the study (UMIN trial ID: UMIN000045941, first registered 01/11/2021). This

report is from "The effectiveness of using an aroma seal for stress relief when wearing a mask: a parallel group comparison test" of the study. All study methods were performed in accordance with the relevant guidelines and regulations.

## Participants

In this study, we recruited 67 healthy university students aged 18 years or older who understood the study's purpose and content, provided written consent, and volunteered to participate. Eligible participants were those who, to the extent that it did not impede the conduct of the study, had the ability to comprehend the Japanese language, both spoken and written, allowing them to understand the study instructions. We excluded individuals who: (1) declined to participate voluntarily; (2) regularly used aroma essential oils; (3) disliked the scent of aroma essential oils; (4) had known allergies to aroma essential oils; (5) had chronic diseases (such as hypertension or epilepsy); (6) required drug therapy for mental illness; (7) experienced allergic reactions during the patch test; (8) did not report stress from wearing masks initially; or (9) were deemed ineligible by the principal investigator. After applying these exclusion criteria, a total of 62 participants were included in the study.

## Randomization and masking

Participants were randomly assigned to one of the two groups (intervention or control) in a 1:1 ratio, and were double-blinded to ensure that the participants and data analysts in the study could not discriminate between the groups of participants. The permuted block method with block sizes of 2 and 4 was used for randomization assignment. An independent allocation coordinator prepared the aroma and placebo seals, creating an allocation table for each participant, including unique identification codes. Random allocation was conducted by a third party not involved in the study, and allocation results were securely stored in a locked safe. Blinding of intervention providers and data analysts was maintained until the completion of data analysis. Randomization was performed using the R statistical software(version 4.0.2, Foundation for Statistical Computing, Vienna, Austria).

## Procedures

The baseline period in our study refers to the two-week period immediately preceding the intervention period, during which participants applied the seals to their masks. The baseline values used in our analysis were obtained from the questionnaire responses collected on the day before the intervention started.

All enrolled participants had their health status checked, followed by a skin allergy test (patch test) with aroma oil; only those who did not have allergic reactions were included in the study. The participants were randomly divided into two groups. After a one-week screening period, participants in both groups used seals when they felt stressed while wearing masks during the two-week intervention period. The seal was applied once a day while going out.

In this study, all participants were initially administered a questionnaire survey to confirm their preferred aromas, choosing from aroma categories such as Citrus, Floral, Trees, Oriental, Herb, Spices, and Resin. This assessment, which allowed participants to select multiple preferred aromas, was conducted to confirm comparability by ensuring that there were no differences in aroma preferences between the two groups after randomization. Orange–lime essential oil was adopted for the aroma used in the seal due to its relaxing and refreshing effect. To ensure participants' comfort, participants themselves were allowed to make adjustments, such as cutting the seal with scissors to make it smaller or placing it further from the nose if they found the fragrance too strong.

The responses to the questionnaires were tabulated at three time points: before the implementation of the study, during the intervention period, and at the end of the study. Specifically, the Google Form link for the survey was sent to all participants at noon on the day of tabulation, with responses required by the end of the day. Utilizing Google Forms for the survey ensured that there was no direct contact between the researchers and the participants throughout the study. The recruitment process was completed within a week, by November 10, 2021, and the follow-up was finalized by November 20, 2021. The intervention began within the first week after all participants were randomized. The initial two weeks served as the baseline period, followed by the two-week intervention period.

## Study outcomes

The primary outcome was the Depression, Anxiety, and Stress Scale-21 (DASS-21) [26, 27], which was used to determine whether daily stress from wearing masks was significantly reduced. The DASS-21 has been validated in multiple studies conducted across diverse cultural and situational contexts around the world during the COVID-19 pandemic [28–34]. The secondary outcome was the World Health Organization- Five Well-Being Index (WHO-5) [35], a rating scale for mental well-being that was used to determine whether the participants' mental health significantly improved in every period when they felt stressed while wearing a mask. Furthermore, a four-point Likert scale was employed to assess whether the use of aroma seals alleviated breathlessness experienced from wearing masks.

The DASS-21 is a 21-question rating scale comprising a three-factor construct of depression, anxiety, and stress. The responses to the questions were scored as 0 (did not apply to me at all), 1 (applied to me to some degree or some of the time), 2 (applied to me to a considerable degree or a good part of time), and 3 (applied to me very much or most of the time). The participants responded by selecting the option that was closest to their current condition in the last week. The responses to each question (from 0 to 3 points) were tabulated, and the scores for depression, anxiety, and stress for each factor item were calculated and evaluated. In this study, we used the Japanese version of the DASS-21 (Short-form version of the Depression Anxiety Stress Scale) to assess participants' levels of depression, anxiety, and stress [36]. Higher factor scores indicate more-severe depression, anxiety, and stress.

To assess whether they were feeling breathlessness while wearing a mask, participants responded by choosing the option that was closest to their condition in the last two weeks. The questions were scored from 1 to 4 points, and the response options ranged from 1 (not feeling breathlessness) to 4 (feeling breathlessness). In this study, we employed a 4-level rating to evaluate the perception of mask breathless among participants. This approach was considered suitable for our study's specific context and objectives.

The WHO-5 Well-Being Index consists of five questions, with each question scored from 0 to 5 points. The response options ranged from 0 (at no time) to 5 (all of the time). Participants responded by selecting the option that was closest to their condition in the last two weeks. The original items for the five questions were as follows: Question 1 (W1): "I have felt cheerful and in good spirits"; Question 2 (W2): "I have felt calm and relaxed"; Question 3 (W3): "I have felt active and vigorous"; Question 4 (W4): "I woke up feeling fresh and rested"; and Question 5 (W5): "My daily life has been filled with things that interest me." In this study, we used the Japanese translated version of the WHO-5 [37].

## Statistical analysis

Descriptive statistics were used to present the demographic characteristics of the participants. Numerical data (age, number of outings per week, and outcome measures) were presented as

means and standard deviations, whereas categorical data (sex, stress, and favorite aroma) were presented as frequencies and ratio. To determine the improvement in the primary outcome of the scale for stress, changes in DASS-21 scores measured at the time points (weeks 1 and 2) from those measured at the baseline were assessed for each group (aroma- and placebo-seal use group). To determine whether breath comfort while wearing a mask improved, the groups were assessed for improvement in the mask-breathlessness score on week 2. In addition, to assess improvements in mental health by reducing stress caused by wearing a mask, the changes in WHO-5 scores for each group at the time of measurement (week 2) with respect to baseline values were evaluated. To evaluate safety, the number of participants who experienced adverse events and the incidence of these adverse events were recorded for each group. All outcomes were coded using the statistical software SAS (Version 9.4, SAS Institute, Inc., Cary, NC, USA). DASS-21 and mask-breathlessness scores were analyzed using a mixed model for repeated measures (MMRM) with each outcome variable baseline values as covariates. WHO-5 scores were analyzed using a mixed-effects model analysis of covariance, also incorporating each outcome variable baseline values. A p-value < 0.05 for all tests was considered statistically significant. All analyses were conducted using a two-tailed test. To ensure complete data, a Google response form was configured to prevent submissions of incomplete forms, resulting in zero missing values. Except for those who withdrew from the study, all participants completed all outcomes at each time point; therefore, the number of participants for efficacy and safety analyses was the same. Additionally, we calculated Cohen's d effect sizes for stress, depression, and other relevant measures to assess the practical significance of the results. Efficacy assessments were performed on participants who successfully completed all trial sessions, while safety assessments were conducted for those who received at least one intervention.

## Results

### Participants' information

In this study, 67 participants were screened, and 62 were included and randomly assigned to one of two groups: an aroma-seal use group ($n$ = 31) and a placebo-seal use group ($n$ = 31). However, one participant dropped out during the course of the study. Accordingly, 61 participants completed the study, with 31 and 30 participants in in the aroma- and placebo-seal use groups, respectively (Fig 1).

The participants' demographics and baseline characteristics were comparable between the groups (Table 1). The overall mean age was 21.1 ± 1.6 years, and the majority of participants (88.5%) were female. Additionally, the number of weekly outings was similar between the two groups. Participants were asked to select multiple preferred fragrances. Results showed that participants' preferences for all scents were consistent between both groups. The overall average mask-breathlessness score was 2.43 ± 1.01, and many participants responded that wearing a mask was rather breathlessness.

### Assessment of depression, anxiety, and stress using the DASS-21

DASS-21, which assesses depression, anxiety, and stress, was utilized in our study. In its evaluations at baseline, 1 week, and 2 week, the scores for the Aroma-Seal Use Group appeared to show a decreasing trend (S1 Table). The changes in the DASS-21 score from baseline are summarized below. Compared to the placebo-seal use group, the aroma-seal use group showed a significant reduction in total score ($p$ = 0.02, effect size: 0.63), depression ($p$ = 0.004, effect size: 0.76), and stress ($p$ = 0.02, effect size: 0.63) score changes from baseline values by the first week of the intervention, and in total score ($p$ = 0.04, effect size: 0.54) and depression ($p$ = 0.04, effect size: 0.54) score changes by the second week of the intervention (Table 2).

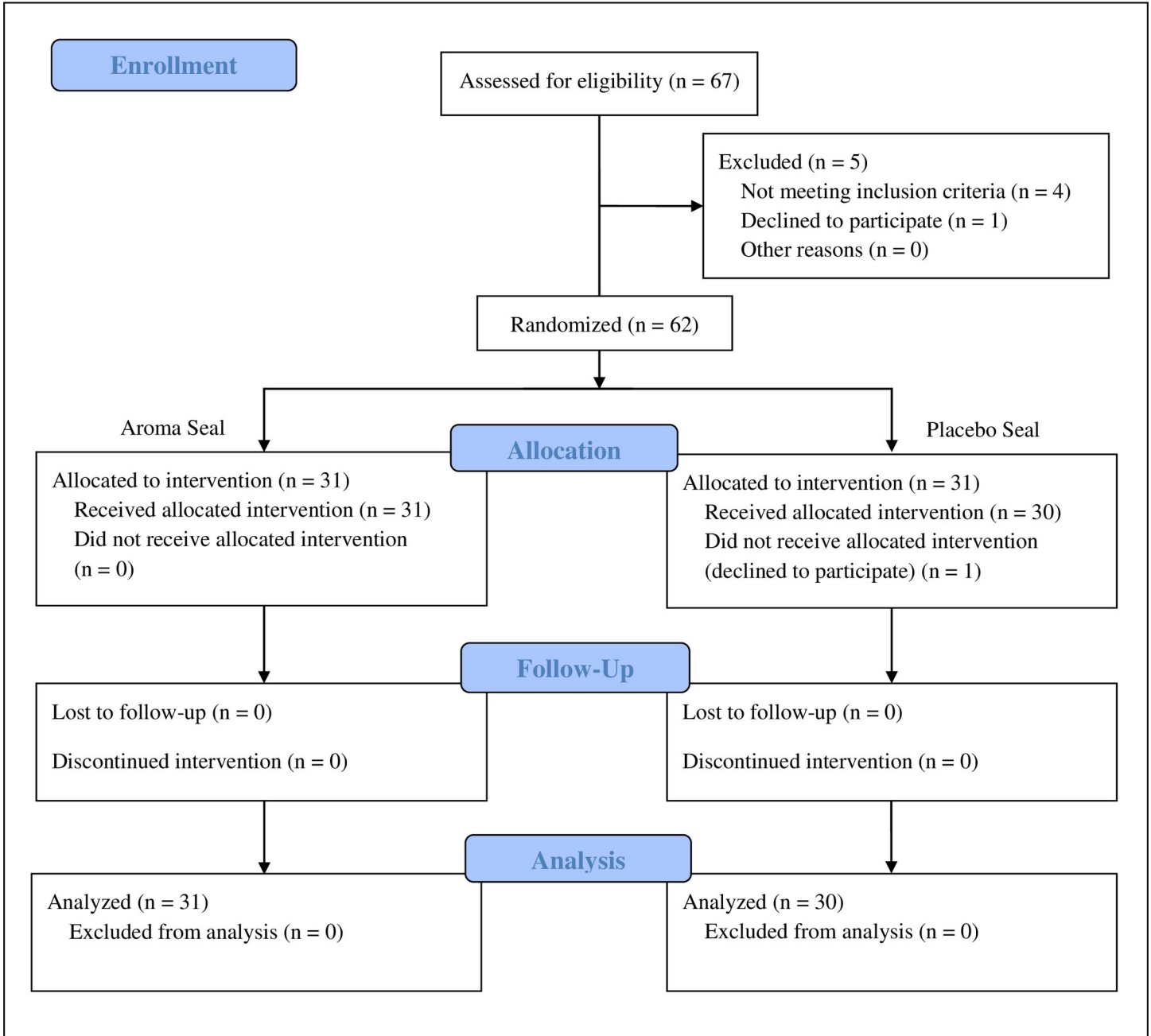

**Fig 1. Flow chart of participant allocation.**

## Improvement in discomfort associated with wearing masks

The aroma-seal use group had mean breathlessness scores of 2.48 ± 1.09, 1.74 ± 0.89, and 1.84 ± 0.86 at baseline, first week, and second week of the intervention, respectively, indicating that the scores changed from "it's rather breathlessness" to "it's not very breathlessness" in many patients. Conversely, the placebo-seal use group had mean breathlessness scores of 2.37 ± 0.93, 2.17 ± 0.79, and 2.30 ± 0.92 at baseline, week 1, and week 2, respectively, with no significant change in breath comfort when wearing a mask. Furthermore, the aroma-seal use

**Table 1. Demographic and baseline characteristics of study participants in the aroma- and placebo-seal use groups.**

| | Aroma-seal use group (n = 31) | Placebo-seal use group (n = 30) | Total (n = 61) |
|---|---|---|---|
| **Age** | 21.3 ± 1.9 | 21.0 ± 1.2 | 21.1 ± 1.6 |
| **Sex** | | | |
| Men | 4 (12.9%) | 3 (10.0%) | 7 (11.5%) |
| Female | 27 (87.1%) | 27 (90.0%) | 54 (88.5%) |
| **Stress** | | | |
| Feeling no stress | 3 (9.7%) | 5 (16.7%) | 8 (13.1%) |
| Feeling a little stressed | 16 (61.6%) | 17 (56.7%) | 32 (54.1%) |
| Feeling stressed | 11 (35.5%) | 7 (23.3%) | 19 (29.5%) |
| Feeling very stressed | 1 (3.2%) | 1 (3.3%) | 2 (3.3%) |
| **Number of outings per week** | 5.3 ± 1.2 | 5.1 ± 1.4 | 5.2 ± 1.3 |
| **Favorite Aroma** | | | |
| Citrus | 25 (80.7%) | 24 (80.0%) | 49 (80.3%) |
| Floral | 15 (48.4%) | 20 (66.7%) | 35 (57.4%) |
| Trees | 7 (22.6%) | 9 (30.0%) | 16 (26.2%) |
| Oriental | 6 (19.4%) | 6 (20.0%) | 12 (19.7%) |
| Herb | 5 (16.1%) | 8 (26.7%) | 13 (21.3%) |
| Spices | 6 (19.4%) | 8 (26.7%) | 14 (23.0%) |
| Resin | 1 (3.2%) | 2 (6.7%) | 3 (4.9%) |
| **Baseline of outcome measures** | | | |
| **DASS-21** | | | |
| Total Score | 8.00 ± 7.14 | 8.67 ± 7.49 | 8.33 ± 7.26 |
| Depression | 2.94 ± 3.00 | 2.60 ± 2.66 | 2.77 ± 2.82 |
| Anxiety | 1.77 ± 1.89 | 2.03 ± 1.99 | 1.90 ± 1.93 |
| Stress | 3.29 ± 3.51 | 4.03 ± 3.80 | 3.66 ± 3.65 |
| **Mask-Breathlessness** | 2.48 ± 1.09 | 2.37 ± 0.93 | 2.43 ± 1.01 |
| **WHO-5** | | | |
| Total Score | 15.55 ± 4.36 | 15.23 ± 3.68 | 15.39 ± 4.01 |
| W1 | 3.42 ±0.81 | 3.30 ± 0.79 | 3.36 ± 0.80 |
| W2 | 3.23 ± 1.02 | 3.23 ± 0.97 | 3.23 ± 0.99 |
| W3 | 3.00 ± 1.03 | 2.93 ± 1.05 | 2.97 ± 1.03 |
| W4 | 2.77 ± 1.15 | 2.83 ± 1.29 | 2.80 ± 1.21 |
| W5 | 3.13 ± 1.20 | 2.93 ± 1.08 | 3.03 ± 1.14 |

The values in Table 1 for quantitative data represent means ± standard deviations (SD), while qualitative data is presented as frequencies (%). The question about "Favorite Aroma" was answered with multiple choices. The DASS-21 total score is the summation of each DASS-21 sub scores. W1, I have felt cheerful and in good spirit; W2, I have felt calm and relaxed; W3, I have felt active and vigorous; W4, I woke up feeling fresh and rested; W5, My daily life has been filled with things that interest me

group showed a significantly lower 0.5-point difference in mean breathlessness score change from baseline than the placebo-seal use group in the one- and two-week time points ($p = 0.004$, effect size: 0.78 and 0.77, respectively; Table 3).

## Assessment of mental well-being using the WHO-5 rating scale

The average score for each item of the WHO-5 was observed to be higher in the intervention group compared to the placebo group (S2 Table). To determine whether the use of aroma seals affected the participants' mental health status, changes in WHO-5 scores from baseline were assessed using scores from the second week of the intervention. There was a statistically

**Table 2. Comparison of DASS-21 score changes from baseline in the aroma- and placebo-seal use groups.**

| | Adjusted Means (95% CI)[a] | | | | 1 week | | | 2 week | | |
| | Aroma-seal use group (n = 31) | | Placebo-seal use group (n = 30) | | Differences of LSMD | p-value | Effect size | Differences of LSMD | p-value | Effect size |
| | 1 week | 2 week | 1 week | 2 week | | | | | | |
| **Total Score** | −4.00 (−7.00 to −1.30) | −3.68 (−5.42 to −1.93) | 0.70 (−2.04 to 3.44) | −1.06 (−2.84 to 0.71) | −4.70 (−8.55 to −0.86) | 0.02 | 0.63 | −2.61 (−5.10 to −0.12) | 0.04 | 0.54 |
| **Depression** | −1.69 (−2.66 to −0.72) | −1.21 (−1.91 to −0.50) | 0.38 (−0.61 to 1.37) | −0.15 (−0.87 to 0.58) | −2.07 (−3.46 to −0.68) | 0.004 | 0.76 | −1.05 (−2.06 to −0.04) | 0.04 | 0.54 |
| **Anxiety** | −0.63 (−1.62 to 0.36) | −0.86 (−1.39 to −0.32) | 0.32 (−0.68 to 1.32) | −0.25 (−0.79 to 0.30) | −0.95 (−2.36 to 0.46) | 0.18 | 0.35 | −0.61 (−1.37 to 0.15) | 0.12 | 0.41 |
| **Stress** | −1.72 (−2.73 to −0.71) | −1.66 (−2.50 to −0.82) | 0.05 (−0.98 to 1.08) | −0.62 (−1.48 to 0.23) | −1.77 (−3.21 to −0.32) | 0.02 | 0.63 | −1.04 (−2.24 to 0.17) | 0.09 | 0.44 |

LSMD, least squares mean difference

significant improvement in W2 (spent in calm, relaxed mood) in the aroma-seal use group compared to the placebo-seal use group (*p* = 0.02, effect size: 0.64; Table 4).

## Adverse event reporting

Safety was assessed for the 61 participants. The number of days of seal use during the intervention period was 5.6 ± 1.2 and 5.4 ± 1.3 days per week in the aroma- and placebo-seal use groups, respectively, and all 61 participants completed the two weeks of the intervention period. One adverse event (3.2%) each was reported in the aroma- and placebo-seal use groups (Table 5). Both adverse events were mild and were resolved two days later. Regarding adverse events for both groups, the same symptoms did not reappear when seal use resumed, and there was no causal relationship with seal use.

## Discussion

This study investigated whether mask seals containing orange–lime essential oils could alleviate stress and breathlessness while wearing a mask and improve mental well-being. The findings revealed a significant improvement in stress and depression, as determined by the DASS-21 scores, in the aroma-seal use group compared to the placebo-seal use group, as well as significantly moderated breathlessness with mask use. In addition, the aroma-seal use group scored significantly higher than the placebo-seal use group on the WHO-5 question, "I have felt calm and relaxed."

Based on the findings of this study, aroma seal masks may have utility in various aspects of daily life. For example, they could offer numerous advantages during extended stays on airplanes. Firstly, the aroma's scent has been suggested to reduce stress and induce relaxation,

**Table 3. Comparison of mask-breathlessness score change from baseline in the aroma- and placebo-seal use groups.**

| Mask Breathlessness | Adjusted Means (95% CI) | | 1 week | | | 2 week | | |
| | 1 week | 2 week | Differences of LSMD | p-value | Effect size | Differences of LSMD | p-value | Effect size |
| **Aroma-seal use group (n = 31)** | −0.72 (−0.94 to −0.49) | −0.62 (−0.87 to −0.37) | −0.49 (−0.81 to −0.17) | 0.004 | 0.78 | −0.53 (−0.88 to −0.18) | 0.004 | 0.77 |
| **Placebo-seal use group (n = 30)** | −0.23 (−0.46 to 0.004) | −0.09 (−0.34 to 0.16) | | | | | | |

LSMD, least squares mean difference

**Table 4. Comparison of WHO-5 score changes from baseline at the end of the two-week intervention in the aroma- and placebo-seal groups.**

|  | Adjusted Means (95% CI) | | Differences of LSMD | p-value | Effect size |
|---|---|---|---|---|---|
|  | Aroma-seal use group (n = 31) | Placebo-seal use group (n = 30) |  |  |  |
| Total Score | 3.32 (2.15 to 4.49) | 1.67 (0.48 to 2.85) | 1.66 (−0.01 to 3.33) | 0.05 | 0.51 |
| W1 | 0.42 (0.12 to 0.71) | 0.20 (−0.10 to 0.50) | 0.22 (−0.20 to 0.64) | 0.3 | 0.27 |
| W2 | 0.80 (0.56 to 1.05) | 0.37 (0.12 to 0.62) | 0.44 (0.09 to 0.79) | 0.02 | 0.64 |
| W3 | 0.60 (0.27 to 0.93) | 0.28 (−0.06 to 0.62) | 0.32 (−0.16 to 0.79) | 0.19 | 0.34 |
| W4 | 0.92 (0.52 to 1.32) | 0.55 (0.15 to 0.95) | 0.37 (−0.19 to 0.94) | 0.19 | 0.34 |
| W5 | 0.60 (0.28 to 0.92) | 0.25 (−0.08 to 0.58) | 0.35 (−0.11 to 0.81) | 0.13 | 0.40 |

LSMD, least squares mean difference

potentially aiding in relaxation and stress reduction during long flights. Additionally, aroma seal masks may enhance comfort and improve sleep quality, making them a potential contributor to passenger comfort and well-being in situations such as air travel. With these possibilities in mind, there is potential to enhance the quality of life for many individuals.

Previous studies have shown that essential oils are effective and safe as fragrances and that inhaling them has psychophysiological effects on humans [38]. The DASS-21, which was used in this study, is a useful measure of depression and anxiety that is frequently used in clinical trials due to its simplicity, brevity, and ability to investigate stress along with depression and anxiety symptoms [39–42]. The results revealed that applying an aroma-containing seal to masks alleviated depression and stress after a week's time. Improvement in depression was observed to persist at the second-week time point. This indicates that while wearing a mask, the orange–lime aroma seal can not only relieve stress but also reduce depression, implying a moderating effect on the stress and depressive effects of wearing masks. Notably, from the value of effect size, the strongest effect was on depression. The DASS-21 total score has been widely used to measure to assess overall mental and emotional well-being in numerous studies. It provides a comprehensive evaluation of the levels of depression, anxiety, and stress experienced by individuals [43–45]. The study found that the total DASS-21 scores were significantly lower in the aroma-seal group than the placebo-seal group at the one- and two-week time points, indicating that aroma seals can also improve overall emotions, as assessed through the DASS-21 total score.

While wearing masks, breathlessness was shown to be relieved more effectively in the aroma-seal use group than the placebo-seal use group. In recent years, aroma sprays for masks have become available on the market; however, aroma seals are less likely to volatilize than aroma sprays and their effects are expected to last longer. Because masks are often worn for long periods of time, aroma seals may be more effective in alleviating breathlessness than sprays.

Among the WHO-5 questionnaire items on the Mental Status Assessment Scale, the score on the item 'I have felt calm and relaxed' was statistically significant, consistent with the

**Table 5. A summary of adverse events in the aroma- and placebo-seal use groups.**

| n (%) | Aroma-seal use group (n = 31) | Placebo-seal use group (n = 30) |
|---|---|---|
| Headache | 1 (3.2%) | 0 (0%) |
| Chest Pain | 0 (0%) | 1 (3.1%) |

Adverse events were defined as those occurring between the time the seal was applied and 14 days after the last use of the mask seal.

previous reports [25, 46, 47]. The aroma was consistently associated with these relaxing and calming mental effects, indicating a sustained impact over the past two weeks.

Several studies on the safety of essential oils have recently been conducted [48, 49]. In this study, there were no adverse events associated with the use of aroma seals because they did not adhere directly to the skin. Furthermore, because it was assumed that participants would have differing scent intensity preferences [50], the participants could cut aroma seals with scissors to adjust them to their individual preference. This may have prevented participants from being overburdened by the intensity of the aroma.

This study has some limitations. First, there was a gender bias among the participants because nearly 90% of the study participants were female. Women are generally known to have a better sense of smell than men [51], which should be considered when interpreting the results. Secondly, it is important to note that the mean age of the participants in this study was 21 years. Therefore, caution should be exercised when interpreting the results, particularly regarding scent preferences, as these preferences may vary among different generations. The findings may not necessarily generalize to individuals from other age groups or demographic backgrounds. In this study, a questionnaire survey about aroma preference was administered to all participants in advance, and the test was conducted after confirming that approximately 80% of the participants preferred citrus. This could explain the positive results obtained. In this study, only citrus scents were used, but using each person's preferred aroma may have a greater effect. Further research on using aroma sprays and comparing their outcomes with those of aroma seals can help in effectively evaluating the efficacy and safety of aroma seals. Additionally, it is important to acknowledge that this study did not pre-calculate the sample size, relying on a convenience sample, which may limit statistical power to detect small effects. This limitation should be considered when interpreting the results and their generalizability. In this study, we employed a simple 4-level rating system to assess mask breath comfort. Given the direct nature of the assessment in our study's context, we believe it was suitable for our study's specific objectives.

The significance of this research is that it was conducted in a situation in which people all over the world were wearing masks due to the spread of COVID-19 and experiencing stress and breathlessness. In this study, we conducted a randomized controlled trial using citrus aroma seals to alleviate stress and breathlessness caused by wearing masks. So far, there have been no reports of such a study being conducted on healthy subjects in their daily lives. Masks have traditionally been used to prevent conditions like influenza [52, 53] and hay fever [54], and it's conceivable that mask-wearing might continue to be recommended even after the conclusion of the COVID-19 pandemic. In considering future research avenues, we note the relevance of a recent study that employed functional near-infrared spectroscopy (fNIRS) to compare brain activation patterns during olfactory stimuli in recovered COVID-19 patients and healthy controls [55]. This investigation highlighted the potential neurological implications of COVID-19 infection on olfactory function. Drawing inspiration from their methodology, future research could extend the application of fNIRS or similar functional brain scan techniques to examine the neural responses associated with the use of aroma seal masks.

Ambient scents have long been recognized to influence people's moods and social, emotional, and cognitive well-being [56, 57]. This study, which was conducted to address everyday issues such as mental stress and breathlessness caused by wearing masks, may indicate that the use of aroma seals, which can be carried out using simple and natural procedures, has the potential to be effective in alleviating daily stress in healthy people worldwide.

## Conclusion

This study demonstrated the efficacy of aroma seals containing orange–lime essential oils in relieving mental stress and improving the comfort of breathing while wearing a mask. Therefore, using aroma seals while wearing masks can be expected to improve mental health and alleviate discomfort in breathing, improving the quality of daily life of healthy people worldwide.

## Supporting information

**S1 Table. DASS-21 scores in the aroma-seal use group and placebo-seal use group at baseline, 1 week, and 2 week.**
(DOCX)

**S2 Table. WHO-5 scores in aroma seal and placebo groups at baseline and 2 week.**
(DOCX)

**S1 Checklist. CONSORT 2010 checklist of information to include when reporting a randomised trial*.**
(DOC)

**S1 File.**
(DOC)

**S2 File.**
(DOC)

**S3 File.**
(XLS)

**S4 File.**
(XLS)

**S5 File.**
(XLS)

## Acknowledgments

The authors would like to thank all the students who participated in this study. We would like to thank Enago (https://www.enago.jp/) for English language editing.

## Author Contributions

**Conceptualization:** Nobuyuki Wakui, Kotoha Ichikawa.

**Data curation:** Nobuyuki Wakui, Aika Okami, Hinako Kagi, Shoko Kawakubo, Chikako Togawa, Raini Matsuoka, Mai Watanabe.

**Formal analysis:** Nobuyuki Wakui, Kotoha Ichikawa.

**Investigation:** Nobuyuki Wakui, Kotoha Ichikawa, Aika Okami, Hinako Kagi, Shoko Kawakubo, Chikako Togawa, Raini Matsuoka, Mai Watanabe.

**Methodology:** Nobuyuki Wakui, Kotoha Ichikawa.

**Project administration:** Nobuyuki Wakui.

**Writing – original draft:** Nobuyuki Wakui, Kotoha Ichikawa.

**Writing – review & editing:** Miho Yamamura, Shunsuke Shirozu, Yuika Tsubota, Yukiko Yoshizawa, Yoshiaki Machida.

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
