## [Decision Letter · Decision Letter 0]

31 Aug 2023

PONE-D-23-19767

Evaluating the effectiveness of applying aroma seals to masks in reducing stress caused by wearing masks: A randomized controlled trial

PLOS ONE

Dear Dr. Wakui,

Thank you for submitting your manuscript to PLOS ONE. After careful consideration, we feel that it has merit but does not fully meet PLOS ONE’s publication criteria as it currently stands. Therefore, we invite you to submit a revised version of the manuscript that addresses the points raised during the review process.

We have received comments from the reviewers and I request to revise the manuscript accordingly and resubmit the manuscript. 

We look forward to receiving your revised manuscript.

Kind regards,

Kamal Sharma

Academic Editor

PLOS ONE

Comments from PLOS Editorial Office: We note that one or more reviewers has recommended that you cite specific previously published works. As always, we recommend that you please review and evaluate the requested works to determine whether they are relevant and should be cited. It is not a requirement to cite these works. We appreciate your attention to this request.

Journal Requirements:

Reviewers' comments:

Reviewer's Responses to Questions

**Comments to the Author**

1. Is the manuscript technically sound, and do the data support the conclusions?

Reviewer #1: Yes

Reviewer #2: No

Reviewer #3: Partly

2. Has the statistical analysis been performed appropriately and rigorously? 

Reviewer #1: Yes

Reviewer #2: No

Reviewer #3: No

3. Have the authors made all data underlying the findings in their manuscript fully available?

Reviewer #1: Yes

Reviewer #2: No

Reviewer #3: Yes

4. Is the manuscript presented in an intelligible fashion and written in standard English?

Reviewer #1: Yes

Reviewer #2: No

Reviewer #3: Yes

5. Review Comments to the Author

Reviewer #1: I have the following comments for the authors to address. I will review this paper again.

1) Under the introduction, please mention the following important finding about facemask from PubMed:

Search PubMed for: There were significantly less Polish respondents who wore face masks (Poles: 35.0%; Chinese: 96.8% p < 0.001). Significantly more Polish respondents reported physical symptoms resembling COVID-19 infection (p < 0.001), recent medical consultation (p < 0.01), recent COVID-19 testing (p < 0.001), and hospitalization (p < 0.01).

2) Under the methods, when descrbing DASS-21, please mention that DASS-21 was validated in the following countries during the COVID-19 pandemics based on the following studies:

China: Immediate Psychological Responses and Associated Factors during the Initial Stage of the 2019 Coronavirus Disease (COVID-19) Epidemic among the General Population in China. Int J Environ Res Public Health. 2020;17(5):1729. Published 2020 Mar 6. doi:10.3390/ijerph17051729

Spain: The Impact of 2019 Coronavirus Disease (COVID-19) Pandemic on Physical and Mental Health: A Comparison between China and Spain. JMIR Form Res. 2021 Apr 22. doi: 10.2196/27818. Epub ahead of print. PMID: 33900933.

The US: The impact of the COVID-19 pandemic on physical and mental health in the two largest economies in the world: a comparison between the United States and China. J Behav Med. 2021 Jun 14:1–19. doi: 10.1007/s10865-021-00237-7. Epub ahead of print. PMID: 34128179; PMCID: PMC8202541.

Poland: The Association Between Physical and Mental Health and Face Mask Use During the COVID-19 Pandemic: A Comparison of Two Countries With Different Views and Practices. Front Psychiatry. 2020;11:569981. Published 2020 Sep 9. doi:10.3389/fpsyt.2020.569981

Iran: https://www.mdpi.com/2673-5318/2/1/6

Philippines: Psychological impact of COVID-19 pandemic in the Philippines. J Affect Disord. 2020 Aug 24;277:379-391. doi: 10.1016/j.jad.2020.08.043. Epub ahead of print. PMID: 32861839.

Vietnam: Evaluating the Psychological Impacts Related to COVID-19 of Vietnamese People Under the First Nationwide Partial lockdown in Vietnam. Front Psychiatry. 2020 Sep 2;11:824. doi: 10.3389/fpsyt.2020.00824. PMID: 32982807; PMCID: PMC7492529.

3) Under discussion, please discuss how the research finding can apply in daily use e.g. wearing aroma seal mask for long time in the aeroplane.

4) Under discussion, I recommend the authors to discuss future research direction based on the methods of the following study to use functiional brain scan to test smell and they can propose a similar study using aroma seal mask verus normal mask:

Comparison of Brain Activation Patterns during Olfactory Stimuli between Recovered COVID-19 Patients and Healthy Controls: A Functional Near-Infrared Spectroscopy (fNIRS) Study. Brain Sciences. 2021; 11(8):968. https://doi.org/10.3390/brainsci11080968

Reviewer #2: Wakui et al. submitted a manuscript titled “Evaluating the effectiveness of applying aroma seals to masks in reducing stress caused by wearing masks: A randomized controlled trial,” evaluating the stress-relieving effects of aroma seals attached to masks using DASS-21 as a primary measure and WHO-5 and mask suffocation as secondary measures. Though the study has focused on an important objective relevant to the current time, the study has major methodological insufficiencies.

Major comments:

1. There is no sample size calculation done/shown in the manuscript or the protocol (attached as a supplementary file). Sample size justification is highly recommended to examine the results’ validity. Without this, the interpretations are invalid.

2. The results could be biased since 80% (49/61) of patients had citrus as a favorite flavor. Hence sub-group analysis for the rest (those with other aromas as favorites) must be examined.

Minor comments:

1. Was the 4-point Likert scale of mask breathability validated? If so, a reference indicating the scale development would be appreciated.

2. Line 113: The baseline period, as mentioned in the first two weeks, does not fully clarify when the baseline measurements were taken.

3. In Line 33, the word ‘incidence’ in terms of suffocation is unsuitable in this context.

Reviewer #3: Comments

The detail information on the randomization method including software and allocation concealment and who conducted the randomization process is to be provided.

For the inclusion criteria, the language criteria (spoken and written) e.g. Japanese is to be added.

Most questionnaires cited were in English, and I assumed that Japanese version was used in the study. As such, the language version of the questionnaires/survey is to be clearly stated. Any translated version used is to be cited.

There was no information on sample size calculation.

Per protocol analysis to be mentioned.

Line 140, what ratio referring to is be clearly stated.

Line 149, the adjusted variables are to be stated apart from baseline.

Line 158-159, there were discrepancies. It was stated ‘30 and 31 participants in the aroma- and placebo-seal use groups’ respectively but Table 1 and Table 5 it was stated ‘’ Aroma-seal use group (n = 31) Placebo-seal use group (n = 30)’ and ‘Aroma-seal use group (n = 31) Placebo-seal use group (n = 32)’ respectively.

Table 1, mean (sd) to be denoted in the table footnote.

Table 2, spacing is to be used to space out between each variable and the results.

Line 183, the statement ‘All analysis adjusted for baseline value, Time, Time and baseline * Time.’’ is to be mentioned in the statistical analyses section.

Table 4 title, the week assessment to be added.

Table 2, 3 and 4, n to be stated.

Data for baseline/first week/second week of intervention of DASS-21, mental well-being etc is to be provided prior displaying ANCOVA analysis output.

Alignment columns for some tables requires adjustment to ensure the letters are not separated apart.

6. PLOS authors have the option to publish the peer review history of their article (what does this mean?). If published, this will include your full peer review and any attached files.

Reviewer #1: No

Reviewer #2: No

Reviewer #3: No

---

## [Author Response · Author response to Decision Letter 0]

4 Oct 2023

Dear Editors and Reviewers,

We sincerely thank you for reviewing our manuscript. We have addressed the points you raised and made the necessary revisions. The details of the changes are documented in the "Response to Reviewers" file submitted through the system. We kindly ask you to review it. If there is any misinterpretation or the revisions are not satisfactory, please let us know, and we will address them promptly. Thank you very much for your understanding and assistance.

---

## [Decision Letter · Decision Letter 1]

31 Oct 2023

Evaluating the effectiveness of applying aroma seals to masks in reducing stress caused by wearing masks: A randomized controlled trial

PONE-D-23-19767R1

Dear Dr. Wakui,

We’re pleased to inform you that your manuscript has been judged scientifically suitable for publication and will be formally accepted for publication once it meets all outstanding technical requirements.

Kind regards,

Kamal Sharma

Academic Editor

PLOS ONE

Additional Editor Comments (optional):

Hello,

Thanks for revising the manuscript as per the points raised by the reviewers.

Thanks

Reviewers' comments:

Reviewer's Responses to Questions

**Comments to the Author**

1. If the authors have adequately addressed your comments raised in a previous round of review and you feel that this manuscript is now acceptable for publication, you may indicate that here to bypass the “Comments to the Author” section, enter your conflict of interest statement in the “Confidential to Editor” section, and submit your "Accept" recommendation.

Reviewer #1: All comments have been addressed

Reviewer #3: (No Response)

2. Is the manuscript technically sound, and do the data support the conclusions?

Reviewer #1: Yes

Reviewer #3: Partly

3. Has the statistical analysis been performed appropriately and rigorously? 

Reviewer #1: Yes

Reviewer #3: Yes

4. Have the authors made all data underlying the findings in their manuscript fully available?

Reviewer #1: Yes

Reviewer #3: Yes

5. Is the manuscript presented in an intelligible fashion and written in standard English?

Reviewer #1: Yes

Reviewer #3: Yes

6. Review Comments to the Author

Reviewer #1: I am satisfied with the amendments. Please allow for publication for Evaluating the effectiveness of applying aroma seals to masks in reducing stress

caused by wearing masks: A randomized controlled trial

Reviewer #3: The authors have addressed majority of the comments.

Minor comment

Mixed model with repeated measures (MMRM) to be written as Mixed model for Repeated Measures (MMRM).

7. PLOS authors have the option to publish the peer review history of their article (what does this mean?). If published, this will include your full peer review and any attached files.

Reviewer #1: No

Reviewer #3: No

---

## [Editor Report · Acceptance letter]

7 Nov 2023

PONE-D-23-19767R1 

Evaluating the effectiveness of applying aroma seals to masks in reducing stress caused by wearing masks: A randomized controlled trial 

Dear Dr. Wakui:

I'm pleased to inform you that your manuscript has been deemed suitable for publication in PLOS ONE. Congratulations! Your manuscript is now with our production department. 

Kind regards, 

on behalf of

Dr. Kamal Sharma 

Academic Editor

PLOS ONE